# Maggot Mass Effect on the Development and Survival of Forensically Important Blow Flies

**DOI:** 10.3390/insects16070660

**Published:** 2025-06-25

**Authors:** Akomavo Fabrice Gbenonsi, Leon Higley

**Affiliations:** School of Natural Resources, University of Nebraska-Lincoln, Lincoln, NE 68583, USA; gbenonsifab@yahoo.com

**Keywords:** Calliphoridae, *Lucilia sericata*, *Calliphora vicina*, developmental model

## Abstract

Forensic investigators use fly larvae to estimate the time of death in legal cases, relying on the larvae’s characteristics at crime scenes. The development of these necrophagous insects is influenced by environmental temperatures, but when larvae aggregate, they generate significant metabolic heat. Our study aimed to assess whether mass temperature could be a reliable indicator of insect development. We reared larvae from two common blow fly species, *Lucilia sericata* and *Calliphora vicina*, under four densities of the insects themselves. Findings revealed that larger groups developed more slowly, primarily due to space limitations, even with sufficient food. A density of around one thousand larvae was identified to achieve a balance between warmth production and competition. In addition, *L. sericata* showed greater tolerance to higher temperatures than *C. vicina*, which survived less in warmer environments. Our results indicate that relying solely on surface temperature may lead to overestimations of developmental stages, potentially impacting court decisions. We recommend using larger containers and adjusting development calculations based on density to improve forensic estimations of time of death, particularly as climate changes continue to affect these dynamics.

## 1. Introduction

In the context of the legal investigation of a victim’s death, forensic entomology relies on the examination of insects and other arthropods for the estimation of postmortem interval [1]. Two principal methodologies are often used in estimating the PMI: ecological succession and Accumulated Degree Days or Hours (ADD/ADH). The ecological succession approach uses predictable changes in insect species that occur within an ecosystem over the decomposition period. This allows forensic entomologists to establish a chronological sequence of insect colonization based on the geographical and seasonal location of a cadaver [2]. The Accumulated Degree Days (ADD) and Accumulated Degree Hours (ADH) methods use insects’ physiological responses to temperature changes to quantitatively measure heat accumulated units, and this helps in terms of making more precise PMI estimates. Moreover, integrating ecological succession patterns with quantitative ADD/ADH calculations can enhance the accuracy of PMI assessments [3,4,5].

Developmental data serve as a dependable tool for estimating the PMI. This method leverages information obtained from studies on carrion insect species. A crucial aspect of this approach is determining the minimum developmental temperature, often called the base temperature, which is important in calculating the Accumulated Degree Days/Hours (ADD/ADH). The variability of the base temperature can sometimes be substantial, particularly among entomologists, and often depends on the specific species under investigation and the geographic region of discovery [6]. Previous studies on blow flies have demonstrated that temperature affects the rate of insect development. Higher temperatures (to a limit) accelerate the development of immature stages, shortening the insect life cycle, while cooler temperatures prolong it [7].

Research has indicated that adult blow flies had lower survival rates when exposed to high temperatures. Studies on *P. regina* and *L. sericata* show that survival significantly declines above 37 °C, with total death at 44 °C [8]. Likewise, at consistent temperatures of 30 °C, *Calliphora varifrons* do not properly emerge [9]. Although maggot mass formation and heat generation have been documented in research for many years, our knowledge of maggot masses, especially the entire aggregation’s physiological ecology, remains limited. Understanding the potential survival mechanisms of larvae within these aggregations is of great interest for a more precise PMI estimation.

This study aims to analyze the temperature distribution within larval aggregations to determine the optimal mass size, investigate its thermal impact on survival and developmental rates, determine lethal temperatures, and examine the maggot mass effect on the developmental processes of major forensic blowflies.

## 2. Materials and Methods

### 2.1. Rearing of Blow Fly Colonies

*Calliphora vicina* and *Lucilia sericata* (Diptera: Calliphoridae) were purchased online from BestBait.com (Marblehead, OH, USA; URL accessed on 2 June 2023), and the species were confirmed by the authors. Five hundred larvae of each species were obtained in containers filled with pine shavings, and upon arrival, the larvae were placed in separate 20 cm × 20 cm × 20 cm net cages for adult emergence. The emerged adults were maintained at a room temperature of 23.3 °C (±1 °C) with ca. 70% RH under a 16:8 (L:D) photoperiod and provided granulated sugar and water *ad libitum*. Raw beef liver was offered as a protein source for oviposition. The infested liver was collected in Petri dishes and distributed in bulk into 1.26 L (ca. 8 × 10 × 16 cm) plastic boxes containing pine shavings. These boxes were placed in incubators at 25 °C (±0.1 °C) to ensure controlled developmental conditions.

### 2.2. Experimental Design

In this study, we designed a split-plot design where three growth chambers served as experimental units, with two different temperatures (25 °C and 30 °C). The temperature was assigned to each chamber as the whole-plot treatment. Within each growth chamber, there was a total of four containers containing larval masses and substrate, serving as the split-plot. Each container was assigned a treatment factor for different larval density levels in the split-plot. Larvae were divided into groups based on a specific number of individuals, respectively, 50, 200, 1000, and 2000. This experiment was repeated three times.

### 2.3. Choice of Larvae

To collect eggs for the experiment, an 18 mL cup containing diced liver was placed inside the adult fly cage to serve as an oviposition substrate. Adults deposited their eggs in crevices and spaces within the liver. The collected eggs were then distributed in bulk into different rearing boxes (ca. 8 × 10 × 16 cm), where additional liver was provided as a food source for the larvae. The boxes were placed in rearing chambers (SMY04-1 DigiTherm^®^ CirKinetics Incubators; TriTech Research, Inc., Los Angeles, CA, USA) maintained at 25 °C. On the third and fourth days, respectively, larvae of *L. sericata* and *C. vicina* were randomly selected for experimentation. Larvae were distributed in their respective experimental boxes: 50 (control), 200 (small), medium (1000), and large (2000). At each density, 10% of larvae were randomly selected to record a baseline to evaluate the weight gained. The balance was tared each time before a new larva was placed on it. To ensure a correct weight result, the larva was left on a Metler analytical balance for about 10 s (until the signal of stable balance was observed). After completing the process, the larvae were introduced back to their corresponding experimental box. Three days after setting the experiments for temperature data, larval weight was individually recorded at 10% of their respective density.

### 2.4. Temperature Data

Each box in the rearing chamber contains a disc containing liver at a density of ca. 6 g/50 larvae. A petri dish containing pine shavings was placed in the middle of the box. The petri dishes were used to favor localized mass with a gradient of aggregation. The pine shavings were changed daily to avoid accumulating waste and CO_2_. At each larval density, temperature data were recorded for 4 days. From the edges to the center of the mass, temperature was recorded at 4 points using a ThermoPro digital thermometer. Data were collected three times a day within a 6-h interval starting at 09:00 h.

### 2.5. Developmental Time

We measured the minimum developmental time by recording the earliest time the first individuals completed development from egg to adult under controlled maggot mass conditions (all larvae aggregated into a single mass). Blow fly eggs were initially laid at a room temperature of 23.3 °C, and the experiments were conducted in rearing chambers set at 25 °C and 30 °C. Pupation occurred in pine shavings. Development was monitored daily, and the minimum developmental time was determined as the earliest recorded adult emergence. Environmental conditions were controlled to minimize external influences on development.

### 2.6. Lethal Temperatures

We used a sous vide container filled with water to conduct our lethal temperature assays. This experiment involved three blow fly species: *L. sericata*, *C. vicina*, and *C. vomitoria*. A single larva from each species was placed in a 0.5 mL vial, secured with a steel wire, and suspended within a sinkable plastic shell. Each trial included four vials as replicates.

For each species, two conditions were tested: preconditioned and regular treatments. In the preconditioned treatment, the water was preheated to 32 °C for 15 or 30 min before proceeding with the lethal temperature assessment. The sous vide container was restarted following preconditioning to determine the temperature threshold leading to larval mortality. The standard treatment served as a control, where larvae were directly subjected to the lethal temperature test without prior preconditioning.

### 2.7. Incubators

The four incubators used in our study were previously described [10,11]. The incubators were customized SMY04-1 DigiTherm^®^ CirKinetics Incubators (TriTech Research, Inc., Los Angeles, CA, USA). These incubators feature microprocessor-controlled temperature regulation, internal lighting, and a recirculating air system to help maintain humidity levels. Unlike conventional growth chambers that rely on coolant and condensers, these units use a thermoelectric heat pump for temperature control. Several modifications were made to enhance functionality, including installing a data port, vertical lighting to ensure uniform illumination across shelves, and an additional internal fan.

According to the manufacturer’s specifications, the incubators maintain an operational temperature range of 10–60 °C with a precision of ±0.1 °C, which is notably more accurate than standard growth chambers. Temperature fluctuations inside growth chambers can differ significantly from programmed settings, whereas internal thermocouple assessments in a replicated study confirmed that temperatures remained within 0.1 °C of the setpoint across all shelves.

### 2.8. Statistical Analysis

All analyses were performed using GraphPad Prism 10.4 for model fitting and visualization, while R version 4.4.3 (R Core Team, 2025) within RStudio 2024.12.1.563 was used for statistical computations. Statistical significance was evaluated at *p* < 0.05 for all tests unless otherwise noted.

For developmental time, pupation success, and emergence rates, Analysis of Variance (ANOVA) was used to assess the significance of chamber temperature, larval density, and their interaction. Data were tested to ensure the validity of assumptions for ANOVA. When ANOVA results were significant, post hoc Tukey tests were conducted to compare specific density groups.

Temperature retention within larval aggregations was analyzed using non-parametric Kruskal–Wallis tests due to non-normal data distributions, followed by Dunn’s tests with Bonferroni correction for multiple comparisons (adjusted *p* < 0.05 considered significant). Weight gain analysis employed a split-plot mixed-effects model to examine fixed effects of density and temperature and their interaction, with random effects accounting for experimental blocks. Model assumptions were verified through residual diagnostics. Significant main effects and interactions were interpreted at *p* < 0.05.

For lethal temperature thresholds, we used two-sample *t*-tests to compare preconditioning treatments, with Welch’s correction applied when variances were unequal. All *p*-values were two-tailed with α = 0.05 determining significance.

Effect sizes were reported for significant findings, including partial η^2^ for ANOVA models and Cohen’s d for *t*-tests, to complement *p*-value interpretations. Data were presented as mean ± SEM.

## 3. Results

### 3.1. Developmental Time

#### 3.1.1. Minimum Developmental Time of *C. vicina*

The ANOVA on the minimum developmental time (in days) of *C. vicina* showed that the effect of chamber temperature alone was not statistically significant (F = 3.125, *p* = 0.096). However, density significantly impacted the minimum developmental time (F = 289.125, *p* < 0.001). Additionally, the significant interaction between chamber temperature and density (F = 9.125, *p* = 0.0009) indicated that the effect of density on developmental time varied depending on the specific temperature conditions (Figure 1).

We found the shortest minimum developmental time of 16 days at a density of 200 larvae when the chamber temperature was set at 30 °C. Furthermore, the minimum developmental time at 30 °C was also lower at a density of 50 larvae compared to when the chamber temperature was at 25 °C. Interestingly, only the aggregation of 2000 larvae resulted in a higher minimum developmental time at a chamber temperature of 30 °C (Figure 1).

The *t*-test for density comparisons showed that larvae reared at higher densities reflect significantly longer developmental times than those at a density of 50 (*p* < 0.001), with a mean difference of 6.5 days. Similarly, the developmental time at densities of 1000 and 2000 was significantly longer than at a density of 200, with a mean difference of 7.33 days (*p* < 0.001). No significant difference in developmental time was observed between the densities of 1000 and 2000 (*p* = 1.00). Similarly, the comparison between the densities of 200 and 50 also showed no significant difference (*p* = 0.098). This suggests that moderate increases in density may not have a substantial impact on developmental time. In contrast, the higher densities of 1000 and 2000 larvae have a more pronounced impact (Figure 2).

#### 3.1.2. Minimum Developmental Time of *L. sericata*

The analysis of variance (ANOVA) conducted on the minimum developmental time of *L. sericata* showed that larval density has a highly significant effect on the minimum developmental time (F = 464.20, *p* < 0.001). In contrast, chamber temperature alone (F = 0.20, *p* = 0.661) and the interaction between chamber temperature and density (F = 0.73, *p* = 0.547) were not significant.

Larvae reared at a density of 1000 and 2000 had significantly longer minimum developmental times than those at 50 (*p* < 0.001), with mean differences of 6.33 and 7.33 days, respectively. The minimum developmental time at densities of 1000 and 2000 was significantly longer than at 200, with mean differences of 6.5 and 7.5 days (*p* < 0.001). However, the difference between densities of 200 and 50 was not significant (*p* = 0.920). This suggests that moderate increases in density may not substantially alter developmental time.

These results suggest that developmental time for *L. sericata* is susceptible to larval density but not to chamber temperature within the tested range. The significant delays in development at high densities may reflect increased competition for resources or stress associated with crowding (Figure 3).

### 3.2. Percentage of Pupae Obtained Across Density and Chamber Temperature

#### 3.2.1. *C. vicina*

The ANOVA on the pupation percentage for *C. vicina* showed that larval density significantly affects pupation success (F = 53.95, *p* < 0.001). In contrast, chamber temperature alone (F = 0.096, *p* = 0.761) and the interaction between chamber temperature and density (F = 0.165, *p* = 0.919) were insignificant in pupation rate.

Larvae reared at higher densities of 1000 and 2000 showed significantly lower pupation rates than those at lower densities. Pupation rates at densities of 1000 and 2000 were substantially lower than at 50, with mean differences of −36.15% (*p* = 0.000005) and −55.21% (*p* < 0.001), respectively. Similarly, pupation rates at densities of 1000 and 2000 were significantly lower than at 200, with mean differences of −22.9% (*p* = 0.0009) and −41.96% (*p* < 0.001), respectively. There was also a significant reduction in pupation rate between densities of 2000 and 1000 (mean difference = −19.06%, *p* = 0.0047), which indicated that increased larval aggregation at higher densities continues to impact pupation success negatively. However, the difference between 200 and 50 was insignificant (*p* = 0.055), suggesting that moderate increases in density may not drastically reduce pupation success (Figure 4).

#### 3.2.2. *L. sericata*

The ANOVA on the percentage of pupation for *L. sericata* shows that larval density significantly affects pupation success (F = 5.07, *p* = 0.012). In contrast, chamber temperature alone (F = 0.103, *p* = 0.753) and the interaction between chamber temperature and density (F = 1.93, *p* = 0.166) were insignificant in pupation rate.

Pupation rates at a density of 2000 were significantly lower than those at 50 (*p* = 0.033), at 200 (*p* = 0.025), and at 1000 (*p* = 0.024), indicating that high larval crowding negatively impacts pupation success. However, no significant differences existed between 50, 200, and 1000 densities (Figure 5).

The ANOVA test on the percentage of emergence for *C. vicina* showed that both chamber temperature (F = 105.18, *p* < 0.001) and larval density (F = 23.75, *p* < 0.001) had highly significant effects on emergence rate. In addition, the interaction between chamber temperature and density was also significant (F = 6.76, *p* = 0.004). This indicates that the effect of density on emergence varies depending on the temperature conditions. This suggests that certain temperature-density combinations may enhance or reduce emergence success, possibly through changes in metabolic rates, competition for resources, or stress levels associated with crowding. Emergence rates at densities of 1000 and 2000 were significantly lower than at 50 and 200. Specifically, emergence at densities of 1000 and 2000 was significantly lower than at 50 by 25.07% (*p* = 0.0003) and 26.61% (*p* = 0.0002), respectively. Similarly, emergence at densities of 1000 and 2000 was also significantly lower than at 200 by 28.58% (*p* = 0.00007) and 30.12% (*p* = 0.00004), respectively. However, the difference between 1000 and 2000 densities was insignificant (*p* = 0.987). The difference between 200 and 50 was also insignificant (*p* = 0.873), indicating that moderate increases in density may not substantially affect emergence success. Emergence at 30 °C was significantly lower than at 25 °C, with a mean difference of 33.70% (*p* < 0.001). This result indicates that higher temperatures negatively impact emergence success (Table 1).

The ANOVA test on the percentage of emergence for *L. sericata* revealed that both chamber temperature (F = 17.35, *p* < 0.001) and larval density (F = 58.85, *p* < 0.001) had highly significant effects on emergence success. Furthermore, the interaction between chamber temperature and density was also significant (F = 5.79, *p* = 0.007). Emergence at densities of 1000 and 2000 was significantly lower than at 50 by −68.35% (*p* < 0.001) and −69.26% (*p* < 0.001), respectively. Similarly, emergence at densities of 1000 and 2000 was significantly lower than at 200 by −47.25% (*p* < 0.001) and −48.16% (*p* < 0.001), respectively. However, the difference between 1000 and 2000 densities was insignificant (*p* = 0.999). Moreover, the difference between 200 and 50 was also significant (*p* = 0.021), indicating that even moderate increases in density can reduce emergence success. Emergence at 30 °C was significantly lower than at 25 °C by 18.85% (*p* = 0.0007) (Table 1).

### 3.3. Larval Aggregation Density on Temperature Retention in Maggot Mass Condition

#### 3.3.1. *C. vicina*

The Kruskal–Wallis test results on *C. vicina* reared at 25 °C indicated a significant difference in temperature across the four densities $\chi_{Kruskal-Wallis}^{2}=712.19;\;p<0.0001.\;$ The effect size estimate suggests a moderate to strong effect, with a confidence interval ranging from 0.50 to 1.00. The violin plot, accompanied by box plots, displayed the distribution of temperatures within each density. The median temperature increases progressively from control to high density. This trend suggests that larger density groups are associated with higher temperatures. Pairwise comparisons using Dunn’s test, with Bonferroni *p*-values, confirmed significant differences between most density groups. The adjusted *p*-values indicated that all pairwise comparisons were highly significant p<0.0001. The most substantial differences are observed between the control (density of 50 larvae) and larger densities, highlighting a consistent and substantial increase in temperature with density (Figure 6).

The Kruskal–Wallis test on *C. vicina* reared at 30 °C results indicated a significant difference in temperature across the four density categories $\chi_{Kruskal-Wallis}^{2}=324.79;\;p<0.0001$. The effect size estimate suggests a moderate to strong effect, with a confidence interval ranging from 0.56 to 1.00. The violin plot and box plots illustrate the temperature distribution within each density. The median temperature increases from 50 to 200 density and remains relatively stable between 1000 and 2000. This pattern suggests that the transition from 50 to 200 is associated with a temperature rise, followed by a plateau effect in larger groups. Pairwise comparisons using Dunn’s test, with Bonferroni *p*-values, confirmed significant differences between most densities. The adjusted *p*-values indicated that only pairwise comparisons between 50 and larger densities were highly significant p<0.0001. The strongest differences were observed between 50 and the other densities (Figure 7).

For *C. vicina* reared at 25 °C, the Kruskal–Wallis test was conducted separately for each density to evaluate whether temperature varied significantly across different positions. The results revealed significant differences in temperature distribution within each density. In the group of 50 larvae, the test indicated a significant effect of position on temperature $\chi^{2}=10.75p=0.0104$. A similar trend was observed in the group of 200, where the chi-square statistic was slightly higher, and the *p*-value was even lower $\chi^{2}=11.62;p=0.000652$. The most substantial effect was detected in the 1000 group category, with a chi-square statistic $\chi^{2}=15.93;p<0.0001$. In contrast, the 2000 category showed the weakest effect $\chi^{2}=6.59;p=0.0103$. While still significant, this result suggests that positional differences in temperature were less pronounced in more prominent individuals compared to the smaller densities. Pairwise comparisons using Dunn’s test with Bonferroni correction revealed significant temperature differences between the edge and center positions across all densities. Respectively, in density, 50 larvae $\left(p=0.0010\right),\;\;$200 larvae (p=0.00065), 1000 larvae $\left(p<0.0001\right)$, and 2000 larvae (p=0.0103). These results confirm that temperature distribution differs significantly between the edge and center positions. The most substantial effect was in the medium-size group and was least pronounced in the large category (Figure 8).

At a rearing temperature of *C. vicina* at 30 °C, the test showed no significant difference across all density groups in temperature generated in the different positions of maggot aggregation. Respectively, in the 50 larvae group, $\left(\chi^{2}=0.003;\;p=0.955\right),\;$200 larvae $\left(\chi^{2}=0.734;p=0.392\right)$ 1000 larvae $\left(\chi^{2}=0.4648;p=0.495\right)$, and 2000 larvae $\left(\chi^{2}=0.544;p=0.460\right)$ were present (Figure 9).

#### 3.3.2. *L. sericata*

The Kruskal–Wallis test conducted on *L. sericata* reared at 25 °C indicated a significant difference in temperature across the four density categories $\chi_{Kruskal-Wallis}^{2}=475.50;\;p<0.0001$. The plot showed that the median temperature increases progressively from control to larger larval aggregation, with a consistent rise across density groups. This pattern suggests that larger densities tend to have slightly higher temperatures. Pairwise comparisons suggested a significant difference between all density groups p<0.0001, with the most substantial differences observed between densities 50 and 2000 (Figure 10).

The Kruskal–Wallis test conducted on *L. sericata* reared at 30 °C revealed a significant difference in temperature among the four densities $\chi_{Kruskal-Wallis}^{2}=324.79;\;p<00001.$ This plot depicted the temperature distribution across density categories. The median temperature increased from 50 to 200 larval groups and remained relatively stable between the density of 1000 and 2000 groups. This trend suggests that the shift from 50 to 200 larval aggregation is associated with a temperature increase, followed by a plateau effect in larger groups. Pairwise comparisons showed significant differences among most density categories. The adjusted *p*-values indicate only comparisons involving the 50 versus 2000 density categories were highly significant p<0.0001 (Figure 11).

The *L. sericata* reared at 25 °C; the Kruskal–Wallis test revealed significant differences in temperature distribution within each density. In the 50 larvae group, the test indicated a significant effect of position on temperature (χ²=4.08, p=0.043). In the 200 aggregations, the chi-square statistic was much higher, and the *p*-value was lower (χ²=18.35, p<0.0001). In the density of 1000 larvae, the strongest effect was detected ( χ²=12.31; p=0.00045)., the effect was weaker for the 2000 larval group (χ^2^ = 7.11, *p* = 0.0077) compared to the smaller categories. Pairwise comparisons using Dunn’s test with Bonferroni correction revealed significant temperature differences between the edge and center positions across all density categories. Respectively, 50 (p=0.043), 200 (p<0.0001), 1000 (p=0.00045), and 2000 (p=0.0077) (Figure 12).

Reared at 30 °C, the test revealed no significant difference across all densities in terms of temperature generated in the different positions of maggot aggregation. Respectively, in the 50 larvae group $(\chi^{2}=0.0566;p=0.954),\;$200 $\left(\chi^{2}=01.1275;p=0.259\right)$, 1000 $\left(\chi^{2}=0.8790;\;p=0.379\right)$, and 2000 $\left(\chi^{2}=0.9919;p=0.3212\right)$ (Figure 13).

### 3.4. Thermal Dynamics of C. vicina Larval Aggregations: Influence of Group Size on Heat Retention

Figure 14 illustrates the relationship between temperature and location within the mass of *C. vicina* across different density categories. There is an increase in the temperature with position in the mass for all groups, following an asymptotic exponential trend. The aggregation of 50 larval groups maintained the lowest temperature across all positions. In contrast, the 200, 1000, and 2000-sized groups exhibited higher temperatures than the 50 larvae. The exponential increase in temperature with position within the mass suggests that heat retention is better toward the center of the aggregation. The fitted model implied that heat accumulation reaches a saturation point in larger masses. The significant thermal differences among density categories suggest that larger aggregations facilitate localized thermogenesis. The control group’s consistently lower temperature further supports the role of larval mass in heat generation. These thermal dynamics could influence larval survival, feeding efficiency, and intra-mass competition.

### 3.5. Thermal Dynamics of L. sericata Larval Aggregations: Influence of Group Density on Heat Retention

The 50 larval groups maintained the lowest temperatures throughout the observed locations through minimal self-generated heat. In contrast, 200, 1000, and 2000 larval aggregations exhibited significantly higher temperatures, with the largest group reaching the highest thermal values. This pattern supports the hypothesis that aggregation facilitates thermoregulation, likely through metabolic heat production and reduced heat dissipation. The non-linear temperature increase observed across locations suggests that larvae in denser clusters experience a thermal gradient, with core individuals benefiting from maximal heat retention (Figure 15).

### 3.6. Temperature Dynamics of C. vicina Across Densities

The temperature response of *C. vicina* over time varied significantly depending on larval density (Figure 16). Initial temperatures remained relatively stable across all densities for the first 40 to 60 h of observation. However, a marked decline in temperature was observed thereafter before stabilizing at lower values around 70 to 100 h. Larger larval densities consistently maintained higher temperatures compared to 1000 and 200 larvae. Specifically, the 2000 larvae group showed the highest temperatures throughout the observation period, followed by the 1000 and 200 groups. This suggests that body size influences thermoregulation or heat retention capacity. This trend is consistent with the expectation that larger larval masses generate more metabolic heat due to increased biomass and collective metabolic activity (Figure 16).

The fitted curves follow a sigmoidal pattern. This model indicates a threshold-dependent transition, where a rapid decline phase follows the initial plateau before stabilizing. Such a pattern suggests that the observed temperature drop could reflect a physiological limit or metabolic exhaustion, possibly related to resource depletion or increased metabolic stress within the larval aggregation. The higher temperatures observed in larger larvae are likely due to increased metabolic heat production associated with greater body mass and aggregation size. The sharp decline phase may signal a shift in the maggot mass’s metabolic state in association with the transition from feeding to migratory behaviors.

### 3.7. Temperature Dynamics of L. sericata Across Densities

The temperature response of *L. sericata* over time followed a quadratic pattern across different density categories (Figure 17). The temperature increased initially, peaked around 40 to 60 h, and then declined toward the end of the observation period. A total of 2000 larvae groups exhibited consistently higher temperatures than groups of 1000 and 200 larvae throughout the observation period. The model indicated that the temperature response follows a parabolic trend over time, where temperature increases to a peak before declining. This pattern likely reflects the combined effects of initial metabolic activity and subsequent post-feeding migratory behavior (Figure 17).

### 3.8. Larval Weight Gain After Three Days Across Density in Maggot Mass Conditions

#### 3.8.1. *C. vicina*

The linear mixed-effects model revealed a significant effect of density on larval weight gain (F = 590.28, *p* < 0.0001), indicating that larvae reared at different densities experienced differences in weight gain. Specifically, compared to density 50, larvae in the 200-density condition gained significantly more weight (t = 2.40, *p* = 0.017). In contrast, those in the 1000 (t = −10.26, *p* < 0.0001) and 2000 (t = −13.32, *p* < 0.0001) density conditions exhibited significantly reduced weight gain. These findings suggest that as larval density increases, competition for resources intensifies, leading to reduced weight gain.

In contrast, chamber temperature alone did not significantly influence weight gain (F = 0.035, *p* = 0.851), as indicated by the non-significant coefficient for the 30 °C treatment (t = 0.49, *p* = 0.623). However, the interaction between chamber temperature and density was highly significant (F = 23.48, *p* < 0.0001). This suggests that the effect of density on weight gain varied depending on the temperature conditions. Despite this significant interaction effect, none of the pairwise interaction terms were individually significant, implying that the differences were not strong enough at specific density levels to drive substantial changes in weight gain at 30 °C compared to 25 °C (Figure 18).

#### 3.8.2. *L. sericata*

The analysis revealed that chamber temperature significantly impacted weight gain (t = 2.446, *p* = 0.0145). Larvae reared at 30 °C gained slightly more weight than those at 25 °C. In addition, density had a strong influence (F = 413.60, *p* < 0.0001). The increasing density generally reduced weight gain. Especially compared to density 50, at 1000 (t = −10.112, *p* < 0.0001) and 2000 (t = −7.421, *p* < 0.0001), the density group exhibited a significant reduction in weight gain. Conversely, larvae in the small-density group showed a slight but significant increase in weight gain (t = 2.541, *p* = 0.0111).

A significant chamber temperature and density interaction (F = 30.53, *p* < 0.0001) indicated that the effect of temperature varied across density levels. The interaction term for 200 larvae (t = −2.081, *p* = 0.0376) and 2000 (t = −3.636, *p* = 0.0003) density at 30 °C was negative and significant. This result suggests that the weight gain advantage observed at lower densities diminished at higher temperatures (Figure 19).

### 3.9. Lethal Temperature of Major Forensic Blow Flies

Preconditioning significantly enhanced the thermal tolerance of *L. sericata* larvae. A difference of 0.75 °C was observed and statistically significant (*p* = 0.011). The increase suggests preconditioning, likely through a heat-hardening mechanism, facilitated a measurable improvement in the larvae’s ability to withstand extreme temperatures. The results indicate that preconditioning influenced the lethal temperature thresholds of *C. vicina* larvae, though with varying degrees of significance. The mean lethal temperature of larvae without preconditioning was 46.4 °C, while those preconditioned for 15 min and 30 min had mean lethal temperatures of 47.0 °C and 47.1 °C, respectively. This suggests that preconditioning led to a slight increase in thermal tolerance.

The statistical comparison between larvae at 32.9 °C and those preconditioned for 15 min (*t*-test, *p* = 0.024) shows a significant difference, indicating that short-term preconditioning enhanced thermal resistance. However, comparisons between the 33 °C and 15-min preconditioning groups (*p* = 0.431) and between the 15-min and 30-min preconditioning groups (*p* = 0.371) did not yield statistically significant differences. This suggests that while initial preconditioning had a notable effect, extending the preconditioning period to 30 min did not significantly enhance thermal tolerance.

The overall comparison of all groups against the 15-min preconditioning group (*p* = 0.170) did not reach statistical significance, implying that while there is a trend toward increased heat tolerance with preconditioning, group variability may have influenced the outcome.

The results indicate that preconditioning had no significant effect on the lethal temperature threshold of *Calliphora vomitoria* larvae. The mean lethal temperature remained constant at 49.0 °C for both the control group (starting at 32.9 °C) and the group preconditioned at 35 °C for 15 min. The standard deviation was also identical in both groups (0.082), suggesting minimal variability in thermal tolerance among individuals.

The statistical comparison between the two groups (*p* = 0.414) confirms no significant difference in heat tolerance due to preconditioning. This suggests that *C. vomitoria* larvae possess an inherently high thermal tolerance or that the specific preconditioning regime used in this experiment did not induce any measurable physiological changes.

## 4. Discussion

This study shows that larval density significantly determines the minimal developing period for *L. sericata* and *C. vicina*. Because of competition and the physiological stress brought on by crowding, the effect of ambient temperature may differ between species. Following the non-linear dynamics suggested, our findings shared a similar framework of density-dependent competition in necrophagous Diptera [12,13]. For both species studied, the increase in larval density resulted in a significant extension of developmental time, especially at high larvae densities. Meanwhile, the developmental times at low-density larvae were not significantly different for either species. This pattern suggests the existence of a density threshold between 200 and 1000 larvae where resource competition transitions from negligible to biologically significant. We can refer to this phenomenon as the “critical mass effect.” This term describes the density threshold at which larvae transition from “solo” dynamics to “group” dynamics. Below this threshold, their development and resource use remain largely unaffected by their neighbors. However, once the threshold is exceeded, emergent, non-linear effects start to appear.

The absence of developmental delays at lower densities indicates that resource availability remains sufficient to support optimal growth rates when competition is minimal. It is well-documented that blow fly larvae exhibit density-dependent feeding plasticity to mitigate intraspecific competition [14]. Larval growth rates remain stable below certain density thresholds due to efficient food partitioning mechanisms.

The threshold at which density effects become significant was between 200 and 1000 larvae. This transition likely reflects multiple interacting factors, including hypoxia-induced metabolic suppression [15,16], proteolytic enzyme inhibition from waste accumulation [17,18], and pathogenic microbial proliferation in crowded environments [19,20]. These mechanisms collectively create a density-dependent bottleneck effect on development that becomes physiologically consequential beyond a critical larval mass.

For *C. vicina*, the significant temperature-density interaction demonstrates that thermal effects on development are contingent upon larval density. The shortest developmental time was observed at 30 °C with a density of 200 larvae, indicating that higher temperatures may accelerate development under moderate densities by increasing metabolic rates [5,21]. However, the extended developmental period at the maximum density at 30 °C might suggest that at extremely high densities, the advantages of higher temperatures are offset by the combined effects of thermal stress and competition. *Lucillia sericata*, on the other hand, showed no discernible relationship between density and temperature. This suggests that developmental time is primarily dictated by density rather than thermal conditions within the tested temperature range. The lack of temperature effects at tested ranges suggests *L. sericata* possesses more remarkable metabolic plasticity, especially its anaerobic capabilities under crowding stress [22]. This species-specific difference could be attributed to varying thermal tolerance mechanisms or differences in metabolic plasticity between the two species. *Lucillia sericata* prefers warm, sunny conditions and exhibits a developmental minimum between 7.5 °C and 10.0 °C, with optimal larval heat emission and growth occurring between 22 °C and 25 °C [11,23]. *Calliphora vicina,* in contrast, favors cooler conditions, with a lower developmental threshold around 1.0 °C and active development beginning at about 13–16 °C, and it develops most efficiently at moderate temperatures near 20–25 °C [24,25].

Studies on blow fly development have shown that while temperature generally modulates insect growth rates, the extent of this effect can vary significantly among species depending on their ecological adaptations [26,27]. The significant growth delay at high densities points to hypoxic conditions limitation as a primary mechanism [28].

Additionally, accumulating metabolic waste products, such as ammonia and CO_2_, may exacerbate physiological stress [29,30]. These factors match the broader framework of density-dependent competition in necrophagous Diptera as larval aggregations face trade-offs between thermoregulatory benefits and resource depletion. While temperatures exceeding 40 °C for maggots in anoxic conditions were documented as critical, our study recorded a maximum of 35 °C in aerated containers. This suggests that thermal stress and anoxic conditions alone may not fully explain the observed developmental delays under our experimental conditions [31].

While developmental time provides crucial insights into growth dynamics, larval weight gain offers a complementary metric for assessing resource allocation and physiological stress under competitive conditions.

Our results demonstrate that larval weight in *C. vicina* showed significant density-dependent patterns, with increased weight gain at low densities but marked reductions at higher densities. Moderate crowding initially enhances growth through cooperative feeding behaviors before reaching a competitive threshold [32]. The temperature-density interaction suggests that thermal conditions modulate competitive outcomes, potentially through temperature-dependent effects on feeding rates and metabolic efficiency. Elevated temperatures increase metabolic demand while accelerating food depletion rates in crowded conditions [33]. The growth response of *L. sericata* differed from that of *C. vicina*, with both temperature and density demonstrating significant effects on larval weight gain. The increased weight observed at 30 °C proved metabolic acceleration as temperatures enhance enzymatic activity and nutrient assimilation rates. However, the negative interaction between temperature and density reveals important constraints on this thermal advantage. The expected temperature-mediated growth enhancement was attenuated at small densities, possibly due to reduced cooperative feeding behaviors that normally optimize resource utilization in moderately crowded conditions. Similarly, at high densities, intensified competition likely offsets the metabolic benefits of warmer temperatures. Our findings extend this model by demonstrating that the thermal modulation of competition follows density-dependent thresholds, with the most pronounced effects occurring at intermediate densities where resource competition becomes significant but before complete resource depletion occurs. The species-specific responses to temperature and density interactions highlight the complexity of larval development under varying environmental conditions. The ability of *L. sericata* to gain weight at higher temperatures suggests more remarkable metabolic plasticity, allowing larvae to adjust their physiology to optimize growth. In contrast, *C. vicina* appears more constrained by density effects, likely due to differences in competitive behavior or inability to adapt to higher temperatures. These findings reinforce the importance of considering intrinsic and extrinsic factors in modeling larval growth and development in forensic and ecological studies.

The density-dependent emergence patterns observed in both species correspond with the Allee effect threshold model [34,35]. However, we observed a 72% reduction in emergence success at 2000 larvae compared to low-density groups. We could, therefore, identify critical larval mass densities where cooperative feeding transitions to detrimental competition in necrophagous systems [36].

The interspecific variation in thermal tolerance between *C. vicina* and *L. sericata* reflects fundamental differences in physiological adaptation. *C. vicina* showed 58% lower emergence at 30 °C compared to 25 °C, consistent with its known thermal sensitivity near the upper physiological limit [19]. This species’ poor performance under combined high-density and temperature conditions demonstrates how ambient conditions can compound thermal stress. In contrast, *L. sericata* maintained more stable emergence rates across temperatures, confirming its broader thermal tolerance range.

The significant interaction between density and temperature challenges conventional developmental models used in forensic applications [37]. These results emphasize how crowding effects can alter temperature-dependent development patterns [38]. Such complex interactions must be accounted for in PMI estimation, as field conditions rarely involve single stressor scenarios [39].

The metabolic activity of blow fly larvae generates substantial heat within aggregations, creating localized temperature increases that significantly impact developmental physiology. Our findings demonstrate a clear positive relationship between density and temperature elevation in both *C. vicina* and *L. sericata*, confirming established principles of larval thermoregulation [18]. This phenomenon results from the combined effects of collective metabolic output and limited heat dissipation in dense aggregations, as documented in forensic entomology studies [12].

Temperature distribution within maggot masses follows characteristic spatial patterns, with maximum temperatures occurring at the center of medium-sized aggregations (200–2000 larvae). This thermal architecture reflects the balance between heat production from larval metabolism and insulation effects from surrounding biomass [38]. However, as aggregation density increases beyond this threshold, enhanced surface area and convective cooling mechanisms serve to stabilize internal temperatures [40].

Notably, at ambient temperatures of 30 °C, we observed limited additional heat accumulation in larger aggregations. This suggests that environmental conditions may constrain the thermoregulatory capacity of larval masses, potentially limiting their ability to maintain optimal developmental temperatures. These findings have important implications for forensic applications, as they demonstrate how microclimate conditions can substantially alter developmental timelines used in postmortem interval estimation.

The species-specific differences observed in thermal regulation align with known ecological adaptations of these forensically important flies; specifically, *C. vicina*, being more temperate in distribution, shows greater sensitivity to thermal extremes than the more cosmopolitan *L. sericata*.

Preconditioning has been widely recognized as a mechanism that enhances thermal tolerance in insects by inducing physiological changes that allow them to withstand extreme temperatures [41,42]. In this study, preconditioning significantly increased the lethal temperature of *L. sericata* larvae, supporting previous findings that short-term exposure to sublethal heat stress can improve heat resistance through mechanisms such as heat shock protein (HSP) expression and cellular repair processes [43]. The observed increase in thermal tolerance supports the idea that short-term acclimation plays a role in insect survival under fluctuating environmental conditions. Furthermore, Monzon et al. [8] demonstrated that adult *L. sericata* could survive at temperatures up to 43 °C, though oviposition success declined at higher temperatures. The present study extends this understanding by showing that larvae of *L. sericata* can tolerate temperatures beyond those lethal to adults, particularly when preconditioned.

In *C. vicina*, preconditioning also influenced thermal tolerance, though the effect appeared less pronounced. The increase in lethal temperature following short-term preconditioning suggests that heat exposure can induce some level of thermotolerance. However, the lack of a strong trend with extended preconditioning time indicates that additional exposure may not enhance heat resistance beyond a certain threshold. Similar findings have been reported in other insect species, where rapid thermal acclimation provides immediate but limited benefits [41]. Given the natural exposure of blow fly larvae to fluctuating environmental temperatures, their ability to develop short-term heat resistance could be advantageous in thermally variable habitats.

In contrast, preconditioning did not appear to influence the lethal temperature of *C. vomitoria* larvae. Among the three species studied, *C. vomitoria* appears to be more heat tolerant up to 49 °C. The consistent thermal tolerance observed across treatments suggests that this species may possess an inherently higher capacity to withstand extreme temperatures. If *C. vomitoria* larvae have evolved to tolerate higher temperatures without preconditioning, this could indicate species-specific differences in heat adaptation strategies.

## 5. Conclusions

Forensic entomologists rely heavily on blow fly ecological succession patterns and temperature-dependent ADD models to estimate PMI in legal investigations. While these methods are well-established, a critical factor often overlooked is the impact of larval aggregation on development. Blow fly larvae naturally form dense masses, generating metabolic heat that can elevate internal temperatures by up to 20 °C above ambient conditions.

This study demonstrates that larval density significantly influences the developmental period of *L. sericata* and *C. vicina.* A critical density threshold was identified at approximately 1000 larvae, representing the optimal “maggot mass” density for studying aggregation effects. Competition becomes biologically significant at this density, leading to extended developmental times. Below this threshold, resource availability remains sufficient for optimal growth, while higher densities result in severe physiological stress and reduced survival.

A key finding is the non-uniform temperature distribution within larval masses, where heat generated by metabolic activity increases from the edges toward the center. Medium-sized masses (200–2000 larvae) exhibit the most pronounced thermal gradient, with the highest temperatures recorded at the core. However, this self-generated heat does not necessarily accelerate development, as the combined effects of crowding and thermal stress can instead delay growth. Our results indicate that using only internal maggot mass temperatures for forensic development models may lead to overestimating growth rates. The adverse effects of competition and waste accumulation can counteract the potential benefits of metabolic heat.

Additionally, species-specific responses indicate varying thermal tolerances: *C. vicina* shows significant developmental delays under high-density, high-temperature conditions, whereas *L. sericata* demonstrates notable metabolic plasticity, maintaining stable growth rates despite crowding. These differences suggest that climate change may disproportionately affect blow fly species, with temperate-adapted flies like *C. vicina* facing more significant challenges in warming environments.

These findings critically impact PMI estimation, as traditional models often assume that maggot mass temperatures accelerate development. Our study demonstrates that crowding effects can undermine the thermal benefits of metabolic heat in maggot masses. In addition, internal mass temperatures should not be used as the sole metric for developmental predictions. Instead, density-dependent competition must be incorporated into forensic models to avoid overestimating growth rates in casework.

Rising temperatures may impact the aggregation dynamics of blow flies in the context of climate change, potentially benefiting heat-tolerant species such as *L. sericata* while disadvantaging thermally sensitive species like *C. vicina*. An increase in the frequency of heatwaves could push larval masses beyond their thermoregulatory limits, leading to higher mortality rates or disrupted development. Additionally, decreased carrion availability might increase larval crowding, further delaying development and complicating forensic estimations.

## Figures and Tables

**Figure 1 insects-16-00660-f001:**
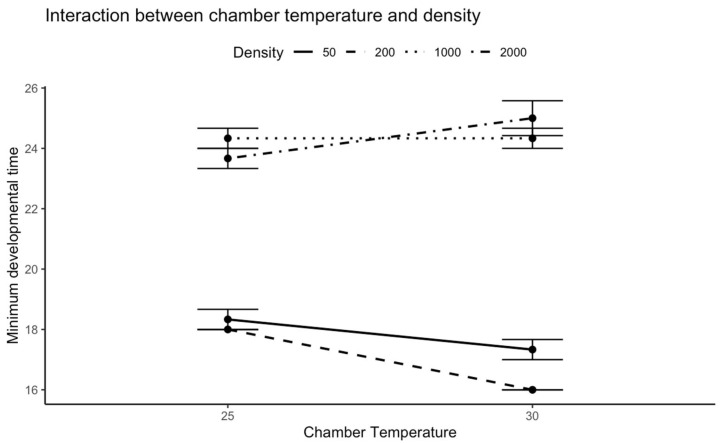
Minimum developmental time (days) of *C. vicina* across chamber temperature and density category.

**Figure 2 insects-16-00660-f002:**
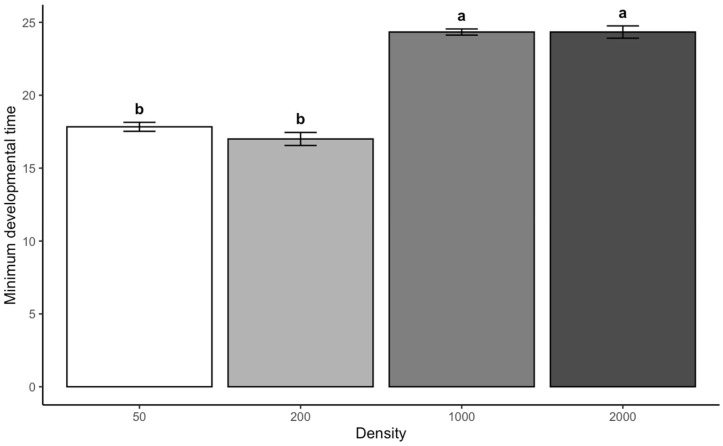
Minimum developmental time (days) of *C. vicina* across density categories. Letters indicate treatment differences at *p* < 0.05.

**Figure 3 insects-16-00660-f003:**
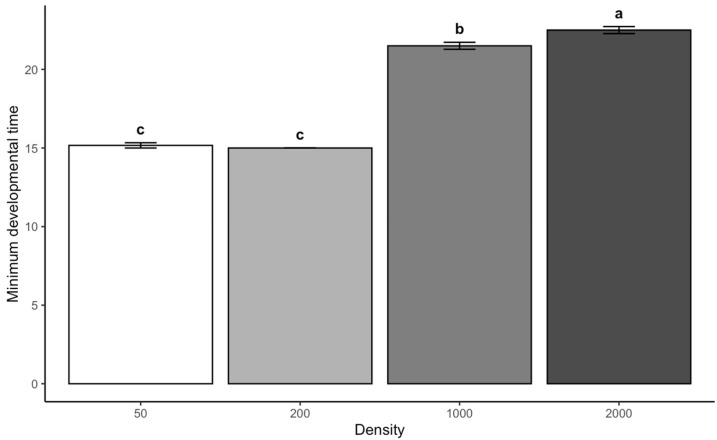
Minimum developmental time (days) of *L. sericata* across density categories. Letters indicate treatment differences at *p* < 0.05.

**Figure 4 insects-16-00660-f004:**
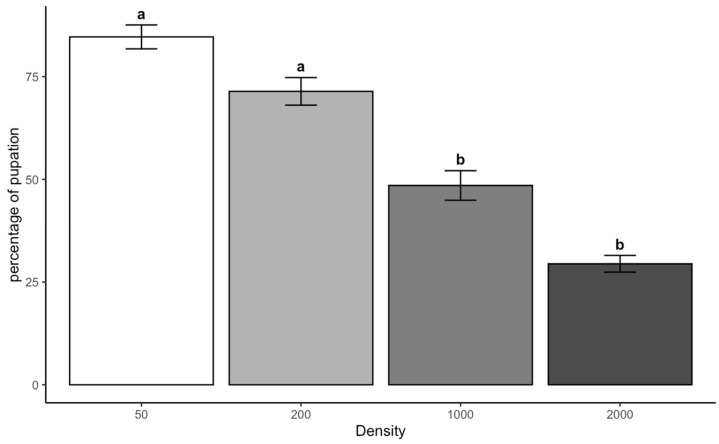
Percentage of pupae of *C. vicina* across the density category. Letters indicate treatment differences at *p* < 0.05.

**Figure 5 insects-16-00660-f005:**
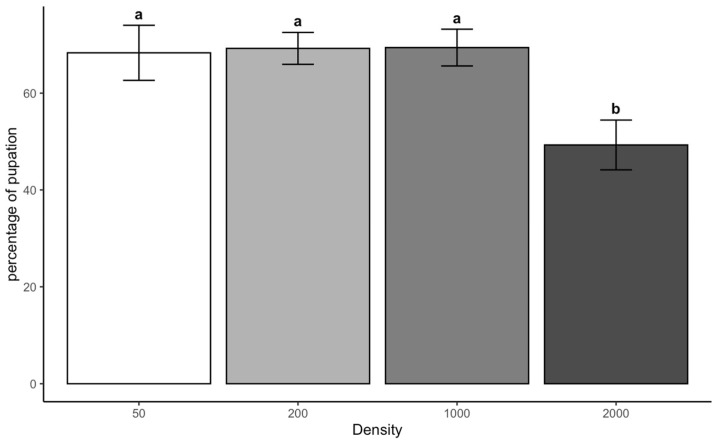
Percentage of pupae of *L. sericata* across the density category. Letters indicate treatment differences at *p* < 0.05.

**Figure 6 insects-16-00660-f006:**
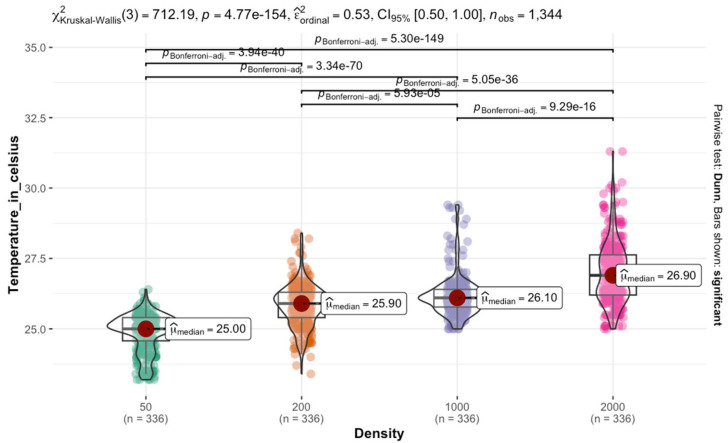
Effect of the aggregation size of *C. vicina* on temperature retention in larval masses at a rearing temperature of 25 °C.

**Figure 7 insects-16-00660-f007:**
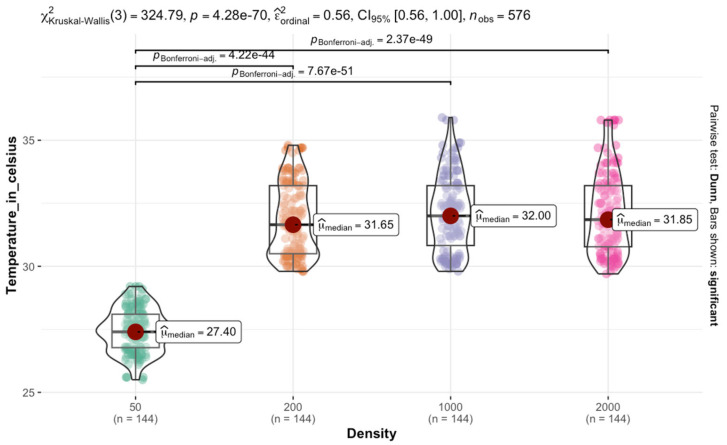
Effect of aggregation size of *C. vicina* on temperature retention in larval masses at a rearing temperature of 30 °C.

**Figure 8 insects-16-00660-f008:**
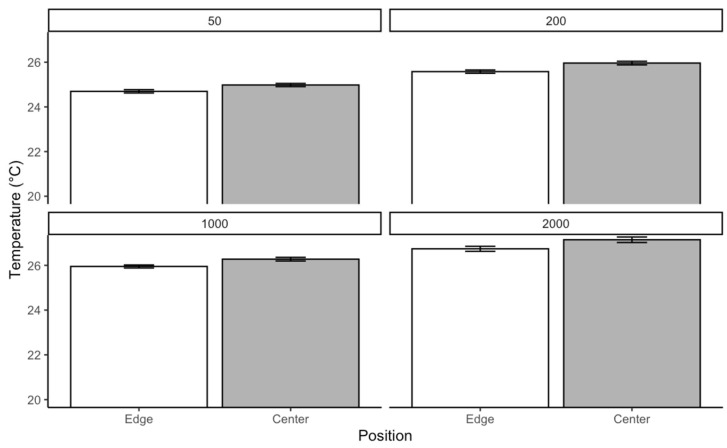
The temperature generated by *C. vicina* within a localized aggregation at a rearing temperature of 25 °C.

**Figure 9 insects-16-00660-f009:**
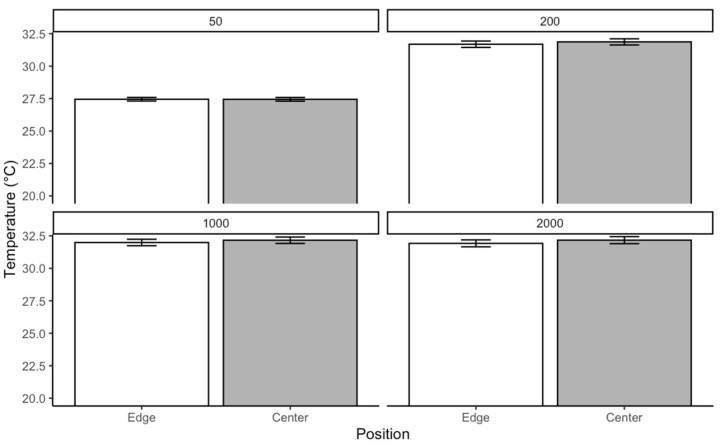
The temperature generated by *C. vicina* within a localized aggregation at a rearing temperature of 30 °C.

**Figure 10 insects-16-00660-f010:**
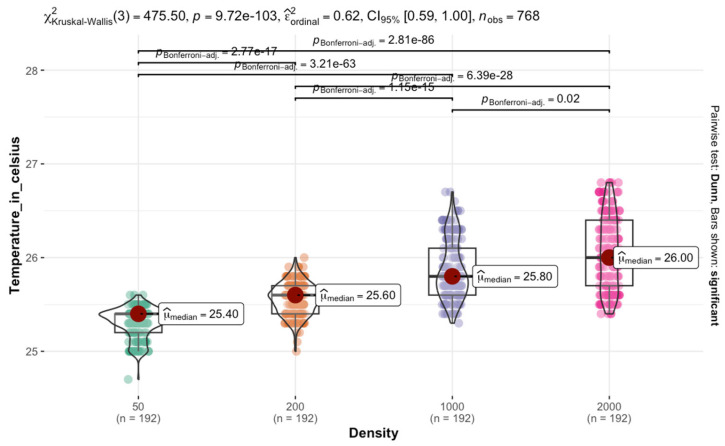
Effect of aggregation size of *L. sericata* on temperature retention in larval masses at a rearing temperature of 25 °C.

**Figure 11 insects-16-00660-f011:**
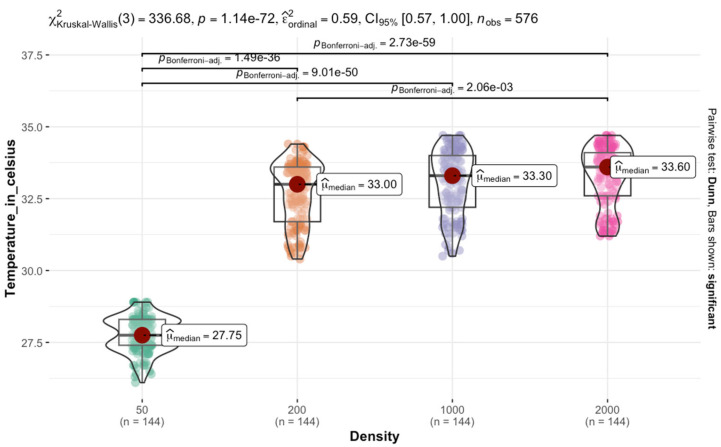
Effect of aggregation density of *L. sericata* on temperature retention in larval masses at a rearing temperature of 30 °C.

**Figure 12 insects-16-00660-f012:**
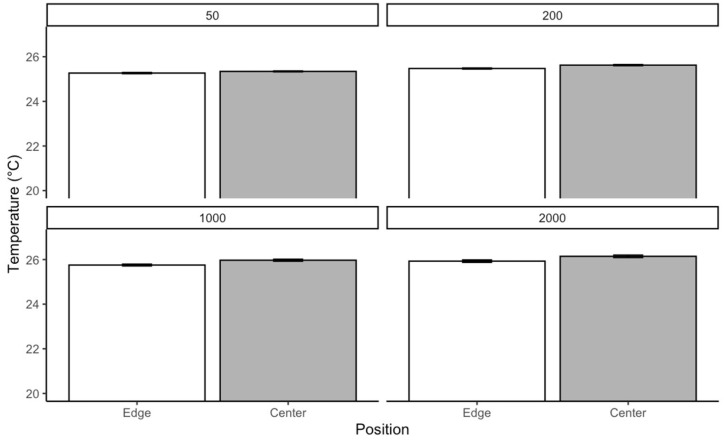
Temperature generated of *L. sericata* within a localized aggregation at a rearing temperature of 25 °C.

**Figure 13 insects-16-00660-f013:**
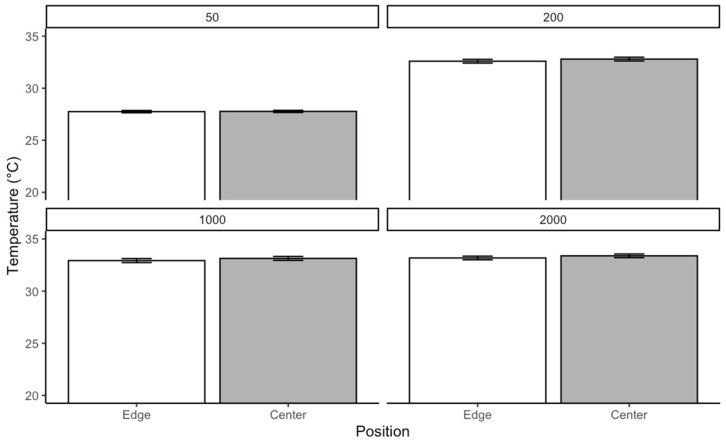
Temperature generated of *L. sericata* within a localized aggregation at a rearing temperature of 30 °C.

**Figure 14 insects-16-00660-f014:**
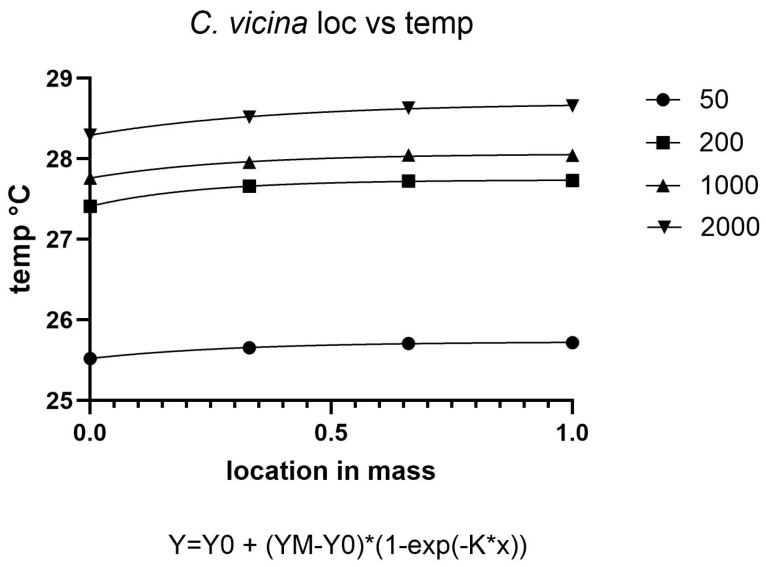
Temperature gradient within *C. vicina* maggot masses, from edge (0) to center (1.0).

**Figure 15 insects-16-00660-f015:**
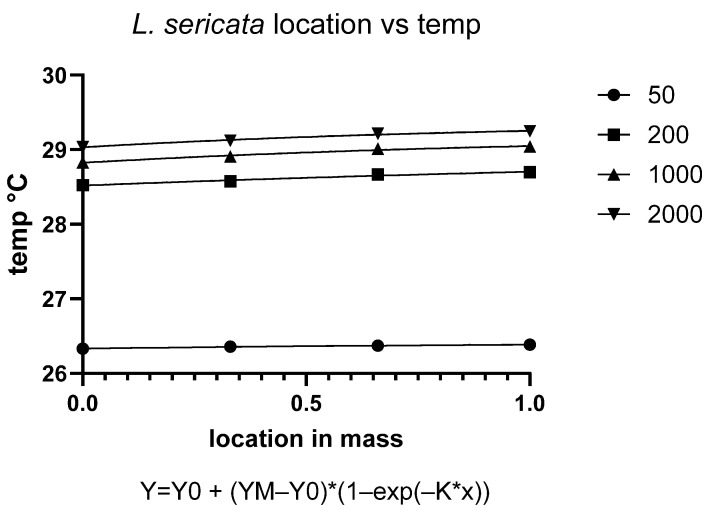
Temperature gradient within *L. sericata* maggot masses from edge (0) to center (1.0).

**Figure 16 insects-16-00660-f016:**
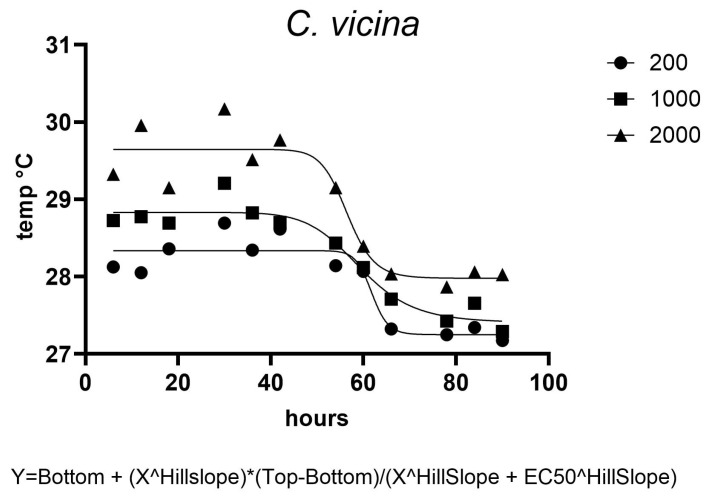
Temperature dynamics of *C. vicina* maggot mass over time at different larval densities.

**Figure 17 insects-16-00660-f017:**
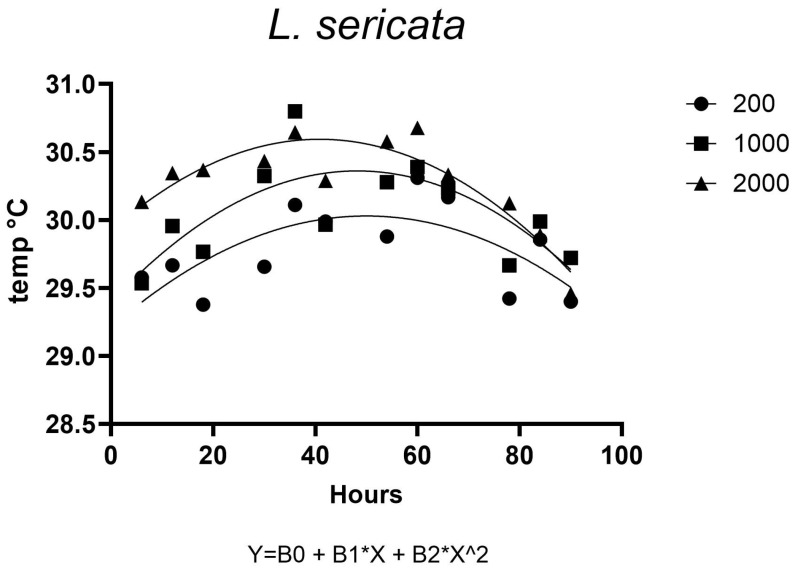
Temperature dynamics of *L. sericata* maggot mass over time at different larval densities.

**Figure 18 insects-16-00660-f018:**
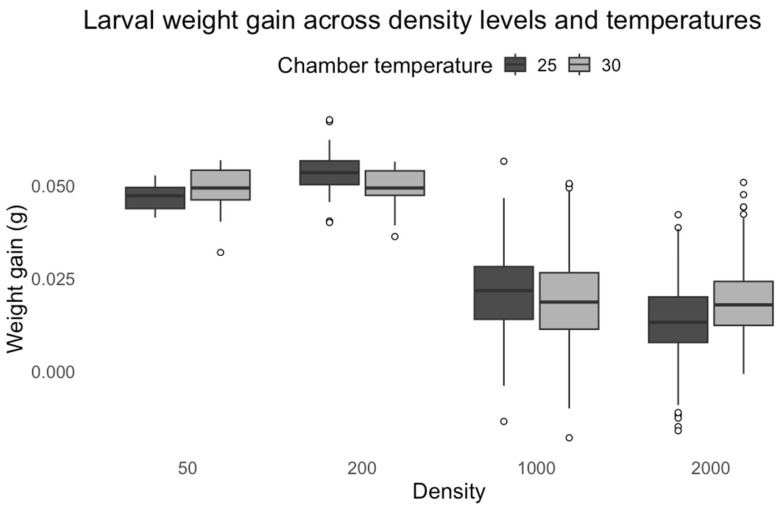
*C. vicina* larval weight gain across density levels and temperatures.

**Figure 19 insects-16-00660-f019:**
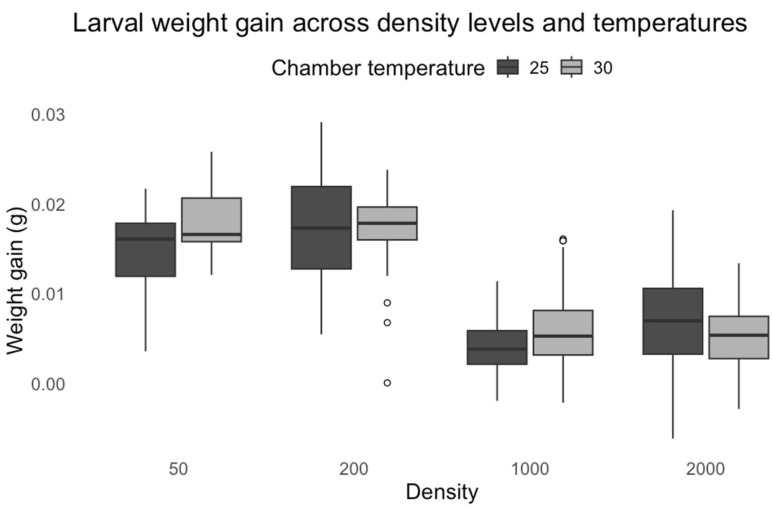
*L. sericata* larval weight gain across density levels and temperatures.

**Table 1 insects-16-00660-t001:** Emergence rate of *C. vicina* and *L. sericata* at various maggot mass densities.

Species	Chamber Temperature	Density	Mean	SE	
*C. vicina*	25	50	58.421785	3.8178312	a
		200	75.789467	5.8479932	a
		1000	29.381405	6.1040901	b
		2000	27.45687	5.0577583	b
	30	50	27.495282	7.0744206	b
		200	17.148823	3.0609373	b
		1000	6.389011	0.9393171	bc
		2000	5.234059	0.8933337	bc
*L. sericata*	25	50	98.420661	0.7976621	a
		200	66.854205	6.4374435	b
		1000	8.379098	0.8725208	c
		2000	5.76173	1.2012799	c
	30	50	51.787701	13.7182281	b
		200	41.153541	9.5875153	b
		1000	5.138126	0.908728	c
		2000	5.928177	1.5892301	c

Letters indicate treatment differences at *p* < 0.05.

## Data Availability

The original contributions presented in this study are included in the article. Further inquiries can be directed to the corresponding author.

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
