# Peer review of "Maggot Mass Effect on the Development and Survival of Forensically Important Blow Flies"

_insects, 2025, doi:10.3390/insects16070660_

Round 1

Reviewer 1 Report

Comments and Suggestions for Authors

This manuscript investigates how larval density and associated temperature changes affect the development and survival of two forensically essential blow fly species, Lucilia sericata and Calliphora vicina. There is several important pieces of information missing, especially in the Materials and Methods section, including formatting issues. In the Results section, there is inconsistency between terminologies (size vs. density), and differences in font type and size were detected. The Results and Discussion sections are lengthy; I suggest making them easier to read by summarizing the content. The Conclusion is also lengthy, and Calliphora vomitoria is missing from this section. Overall, this manuscript is not easy to read, is repetitive, and may cause readers to lose focus. Major revisions are needed to summarize the Results, sharpen the Discussion, and condense the Conclusion. Please refer to the commented PDF for your corrections.

Author Response

This reviewers corrections were indicated in a pdf of the manuscript. We addressed all concerns and made corrections in the manuscript. A key point was changing treatment descriptions from size to density, which was changed throughout the manuscript.

The revised manuscript is attached.

Reviewer 2 Report

Comments and Suggestions for Authors

The article addresses an interesting approach about the effect of maggot mass on the development and survival of two forensic important blowflies. The experimental part is complex  and was very detailed. Despite this, several aspects deserve to be highlighted, to improve this version of the article:

  • in the Abstract and Material and Methods: what was the amount of food provided to the larvae?
  • line 49: geographical ...and seasonal?
  • line 62: up to a limit...
  • line 97: can a larva develop alone?
  • line 112: 10% of the larvae were weighted?
  • lines 127-134: what was the pupation site?
  • it would be better to join Figures 1 and 2;
  • line 246: ... and emergence?
  • remove lines 301-302;
  • line 328: control: 50 larvae? (see line 110), because on line 97: control is represented by 1 larva;
  • line 390: difference;
  • line 412: For the....
  • lines 451-452: better in the Discussion?
  • Figure 14: what is the explanation for the x-axis scale?
  • lines 579-580: "This suggests that C. vomitoria larvae possess an inherently high thermal tolerance"; but on lines 18-19: "L. sericata showed greater tolerance to higher temperatures than C. vicina";
  • line 585: "the effect of ambient temperature": explained much later in the Discussion;
  • line 629: "ecological adaptation": explain better;
  • lines 644-645: exploitative x interference competition in blowflies;
  • line 713: supporting or corroborating?
  • lines 743-744: are results of other articles supporting conclusions?
  • lines 769-770: "if the appropriate carrying capacity is not well selected": explain better;
  • in practice, considering real forensic cases, the expert will rarely collect large samples of larvae from corpses;
  • remove refs 44 and 45.

Author Response

The reviewer corrections listed below were all corrected in the revised manuscript and are indicated through track changes in the revision (which was submitted with reviewer one).

Reviewer 3 Report

Comments and Suggestions for Authors

This study addresses a topic that I have often wondered about in the course of forensic entomology casework. The maggot mass effect has always posed a methodological challenge when rearing fly larvae for experimental or case-related purposes. I truly appreciate the authors for conducting such a timely and important investigation. The study is well-designed and provides valuable insights for both research and practice.
I would be grateful if the authors could consider the following comments and suggestions.

Title: The term ‘forensic blow flies’ might be slightly misleading, as blow flies are not inherently forensic but are rather commonly used in forensic entomology. Consider rephrasing as ‘forensically important blow flies’ or ‘blow flies of forensic relevance’ for clarity.

In Figures: In several figures, the term "size category" is used to distinguish between groups of larvae. Since these categories are based on the number of individuals (i.e., larval density), it may be clearer and more precise to revise this wording to "density category" or "larval density group." This would help avoid potential confusion with individual body size or container dimensions. Additionally, throughout the manuscript (e.g., line 499), the term “size categories” appears to be used to refer to different larval density groups (i.e., 50, 200, 1000, 2000 larvae). Since “size” may be misinterpreted as referring to body size or mass volume, I would recommend revising such instances to “larval density” or “density categories” for terminological consistency and clarity across the manuscript and figures.

For Fig.1: The manuscript includes a detailed figure (Figure 1) showing the interaction between chamber temperature and larval density for C. vicina’s developmental time, which is helpful. However, a similar figure is not presented for L. sericata. While the interaction was not statistically significant, including a comparable plot—even if only to show the lack of interaction—would help readers visually compare the two species and appreciate the contrast in their responses.

Line 50: The transition marker "However" may imply opposition, though the second sentence seems to complement rather than contradict the first. Consider revising it to "Moreover,", "In addition," or "While ADD/ADH methods are effective,..." for clearer rhetorical flow.

Line 80: The authors mention that larvae were obtained from bestbait.com. Given that this supplier is primarily for fishing bait, it may be helpful to clarify whether these larvae represent wild-type populations. Do the authors have reason to believe that the rearing conditions or genetic background of these larvae are comparable to naturally occurring forensic blow flies typically encountered in casework? Additionally, since bait suppliers may not strictly maintain species purity, it would be valuable to mention whether any species confirmation (e.g., morphological or molecular identification) was performed to ensure that only the target species were used in the experiments. This would be especially important for experimental reliability, given the known variability among blow fly species.

Line 87: While the authors mention the use of 1.26 L plastic containers, it would be helpful to include the exact dimensions (e.g., length × width × height) of the rearing containers. Since spatial constraints may influence larval aggregation, heat retention, and competition dynamics, providing precise container measurements would enhance reproducibility and interpretation of the results.

In Figures 6, 7, 10, and 11, the term “aggregation size” is used, but this appears to refer to larval density (i.e., number of larvae per container). For consistency and clarity, it may be better to use “larval density” or “density group” throughout the figures and text, as this is the terminology used in other parts of the manuscript. This would help avoid potential confusion about whether “size” refers to individual larval body size or maggot mass dimensions.

Line 745: In the Conclusion section, the authors identify a density of approximately 1000 larvae as the “optimal maggot mass size.” However, based on the data presented, several developmental outcomes (e.g., shortest developmental time, highest emergence rate, and weight gain) were observed at a density of 200 larvae. At 1000 and 2000 larvae, development was delayed and survival substantially reduced. This seems to suggest that 200 larvae may be closer to the optimal density under the tested conditions. I recommend revisiting this statement to better align the conclusion with the data.

Finally, I wonder whether it might be possible to standardize the appropriate number of fly larvae for rearing not simply based on the total number of individuals, but through a more nuanced approach. For example, could larval density be standardized in relation to the container’s surface area or volume, or to the surface area, volume, or weight of the liver used as a feeding substrate? This might allow for a more reproducible and biologically meaningful metric across studies.

Comments on the Quality of English Language

This manuscript was submitted from a U.S.-based institution, and as a non-native English speaker, I may not be in the best position to fully assess the language. However, I did notice a few expressions that felt awkward or unclear, and I have pointed out some of these in my comments.

Author Response

comment on title: title changed as suggested

comment on size versus density: the terminology for the density treatment has been changed throughout the manuscript including figures to "density" to provide a more precise description

to reduce manuscript length and because the relationship was not significant, we chose not to add a figure for L. sericata

"however" changed to "moreover"

added text indicating that species identity was confirmed by the authors

added container dimensions

"size" changed to "density" throughout

We think density of larvae with a ca. 0.6 g liver/larva food source provides a good indication of how to determine maggot mass size and potential effects. The reviewer's comments are potentially useful for other experimental conditions but numbers with food per larvae strike us as the most generic way to describe maggot density.

Round 2

Reviewer 1 Report

Comments and Suggestions for Authors

The authors have made changes according to the comments.